# Extending Fair Null-Space Projections for Continuous Attributes to Kernel Methods

**Felix Störck** [1]   **Fabian Hinder** [1]   **Barbara Hammer** [1]

## Abstract

With the on-going integration of machine learning systems into the everyday social life of millions the notion of fairness becomes an ever increasing priority in their development. Fairness notions commonly rely on protected attributes to assess potential biases. Here, the majority of literature focuses on discrete setups regarding both target and protected attributes. The literature on continuous attributes especially in conjunction with regression – we refer to this as *continuous fairness* – is scarce. A common strategy is iterative null-space projection which as of now has only been explored for linear models or embeddings such as obtained by a non-linear encoder. We improve on this by extending this to kernel induced feature spaces by means of the "empirical feature space". We theoretically derive this as a direct transformation of the kernel matrix yielding a model and fairness-score agnostic method applicable to continuous protected attributes. We demonstrate that our novel approach in conjunction with Support Vector Regression (SVR) provides competitive or improved performance across multiple datasets in comparison to other contemporary methods.

## 1. Introduction

"The development, deployment and use of AI systems must be fair" is a central statement in the Ethics Guidelines for Trustworthy AI of the EU Commission (2019, p. 12). Similarly fair treatment is in many areas governed by anti-discrimination laws such as Article 21 in the charter of fundamental rights of the European Union (2012) which among others include protection against discrimination based on "age", "sex" and "race". Evidently, in recent years signifi-

cant attention has been drawn to this issue not only in the ML community itself but also by legislators. This led to a series of developments to ensure the fairness of ML systems.

ML systems typically make use of all data provided to them. As a result, they can inherit biases – such as those stemming from ageism or racism – and may learn to reproduce these patterns. Fair ML aims to address this issue by attempting to mitigate or remove such biases. Most commonly, this is done by explicitly marking certain features as protected attributes – for example, "age", "sex" or "race" – which should not influence the learned decision rules, such as determining "creditworthiness" or the "crime probability". The challenge, however, lies in the fact that even if these attributes are not directly included in the training data, they can still be inferred indirectly through hidden correlations, allowing the system to pick up on them unintentionally.

Most often, the focus is on categorical protected attributes (such as "race") in a classification setting. Whilst a large body of literature assumes this setting, an attribute such as "age" is not naturally categorical and other attributes such as "race" are commonly found as a population percentage in a given community which is then often categorized as pre-processing. Further, the more general regression setting is also not as frequently considered thus rendering the available options to ensure the by law required non-discrimination guarantees slim. We will refer to the setup in which the targets as well as the protected attributes are continuous rather than discrete as "*continuous fairness*".

A common strategy in fairness is the idea of "null-space projection": first a direction in space predictive of the protected attribute is determined and then the data projected so that this direction becomes uninformative. An iterative process that each time removes different pieces of information. However, this idea is thus far limited to either linear methods or the application to non-linear embeddings. This work significantly extends the approach to kernel induced feature spaces, specifically our contributions are the following:

- We derive a method to perform a direct transformation of the kernel matrix that corresponds to a null-space projection in the empirical feature space. We show that the theoretical properties of the kernel matrix are

[1]CITEC, Faculty of Technology, Bielefeld University, Germany. Correspondence to: Felix Störck <fstoerck@techfak.uni-bielefeld.de>.

*Proceedings of the 43ʳᵈ International Conference on Machine Learning*, Seoul, South Korea. PMLR 306, 2026. Copyright 2026 by the author(s).

retained and demonstrate out-of-sample extension.

- We illustrate that the transformed kernel is model agnostic: most notably giving rise to a method for fair Support Vector Regression.

- We empirically obtain competitive or improved performance against other contemporary methods on a range of different datasets and fairness scores.

This paper is structured as follows: first we introduce the field of fair machine learning by discussing a selection of related works (Section 2) with a focus on social implications in Section 2.3 before presenting the extension of null-space projections to kernel methods (Section 3). In Section 4 we empirically demonstrate competitive performance on real-world datasets, afterwards discussing our approach, its limitations and directions for future work (Section 5).

## 2. Related Work

We first discuss the general setting of fair ML before diving into specific fairness approaches related to subspace projections and kernel methods.

### 2.1. Fairness in Machine Learning

Two main stream characterizations of fairness in machine learning can be distinguished: *individual* and *group* fairness. The former aims for similar predictions for similar individuals, the latter looks at the group specific predictions given a so-called protected attribute (such as "race" or "age"). For an extensive discussion see Binns (2020). The most common measure for fairness are *Demographic/Statistical Parity* and *Equalized Odds*. The former aims at an equal probability to receive an advantageous decision/label (e.g. credit worthiness) whilst the latter also takes the true label into account. It is not possible to directly compute these for the continuous fairness setting and hence adequate adaptions are required which we discuss next.

### 2.2. Continuous Fairness Measures

In recent years, a set of novel approaches to compute fairness measures for continuous fairness have been introduced – most aim to measure the statistical dependence between the model prediction and the protected attribute. Mary et al. (2019) introduce a fairness measure based on the Hirschfeld–Gebelein–Rényi (HGR) maximum correlation and a method for its estimation based on kernel density estimations. Concurrently, two generalizations of demographic parity were introduced: Giuliani et al. (2023) present (inspired by properties of the HGR) a family of Generalized Disparate Impact indicators (GeDI) and Jiang et al. (2022) put forward "Generalized Demographic Parity" (GDP) by

using a kernel density estimation to weight local and global prediction averages.

A different approach is taken by Narasimhan et al. (2020) by deriving pairwise fairness (PF), a measure based on pairwise comparisons for demographic parity and equalized odds.

It has been mathematically shown (Kleinberg et al., 2017) that the respective objectives of different fairness measures are mutually exclusive. It therefore remains up to the designer of the machine learning system which measure to employ during training. This is aggravated in the continuous fairness setting as computational tractability can become an issue depending on the chosen measure and its parameters.

One strategy is to include one of the fairness measures (or a suitable surrogate) into the optimization problem as a penalty – for example Mary et al. (2019) use a neural network penalized by an upper bound of the HGR (denoted "NN-HGR"), Kong et al. (2025) penalize with a new metric whilst Lai & Guan (2025) rely on an adversarial technique.

### 2.3. Continuous Fairness: Social Implications

This section aims to explore how the wider body of literature including social sciences deal with the issue of continuous fairness and how it is recognized therein.

Stypinska (2023) discusses the general topic of ageism related to AI systems and how this on the technical level can be attributed to under-representation or cutoffs of certain age groups whilst often being neglected in the fairness discourse. Chu et al. (2023) perform an extensive survey to explore the leading sources of age-related biases in AI systems concluding among other things that "the second phenomenon is the misinterpretation of age as a category [...]" (p. 12). The effect of categorizing "age" has been recognized by Hort & Sarro (2022) yet the focus rather remains on finding suitable categories than treating it as a continuous quantity.

Regarding the attribute "race", many works in the social and related sciences commonly work with it as a population percentage, for example when analyzing the connection to fatal police shootings (Zare et al., 2025) or traffic citations (Xu et al., 2024). Note that these works rely on data from the "American Community Survey" which also inspires two of the three datasets used in this work for empirical evaluation.

This demonstrates a clear disparity between how the attributes are dealt with by experts in practice and how they are (often for algorithmic convenience) treated within the ML community.

### 2.4. Sub-space Projections for Fairness

Subspace projections comprise methods aimed at identifying subspaces with specific properties usually to ensure subsequently applied algorithms satisfy certain guarantees.

This is a common approach to ensure workspace constraints in robotics (Dietrich et al., 2015).

In the domain of fairness Pérez-Suay et al. (2017) also introduce a fair dimensionality reduction technique based on the "Hilbert-Schmidt independence criterion" (HSIC) (Gretton et al., 2007).

Tan et al. (2020) take the viewpoint of empirical risk minimization trying to find a subspace of the hypothesis space that fulfills predictive but also fairness properties for which they derive an approach for Gaussian Processes.

Ravfogel et al. (2020) on the other hand suggest to first find a direction in space deemed relevant for predicting the protected attribute – in their case the weights of a linear classifier[1] whose null-space is then used for the projection of either the data or their embedding in a neural network.

This work generalizes the work of Ravfogel et al. (2020) by considering the regression setting for kernel methods i.e. we provide means to work with null-space projections for the non-linear features induced by kernels including cases where the corresponding feature space is infinite dimensional which is a novel contribution. As our approach focuses on kernel methods we discuss previous work on fairness with kernels in the next section.

### 2.5. Fairness in Kernel Methods

There is some work regarding the continuous fairness setting with kernels in the literature: Pérez-Suay et al. (2017) also develop a fair approach for Kernel Ridge Regression (we refer to this as "KRR-FKL") using the HSIC which is extended to Gaussian Processes but without out-of-sample extension (Perez-Suay et al., 2023). Oneto et al. (2020) introduce constraints to the KRR optimization to develop an alternative approach.

Ravfogel et al. (2022) propose a kernelized minimax game for removing non-linear information about categorical[2] protected attributes. Grünewälder & Khaleghi (2021) use the kernel-induced feature space to design features that remain close to the original data while being minimally dependent on the (possibly continuous) protected attributes.

In contrast, even though there exists a variety of different approaches for fair Support Vector Machines (SVMs) (Olfat & Aswani, 2018; Donini et al., 2018; Zafar et al., 2019; Carrizosa et al., 2023) they are neither extendable to regression nor to continuous protected attributes – at least without

---

[1]It is mentioned that this can be extended to the regression setting, however to the best of the authors' knowledge this has not yet been thoroughly investigated in the community.

[2]The formalism itself does not appear to be inherently restricted to categorical protected attributes, but the paper itself as well as the provided implementation exclusively address categorical attributes.

more intricate changes as they mostly rely on modification of the original optimization problem to incorporate different measures of fairness as constraints or penalties.

The method introduced next is model agnostic and applicable for fairness as long as the approach relies on kernels.

## 3. Method

In contrast to other approaches, we do not directly optimize for a specific fairness-score. Rather, we aim to remove information that is required to predict the protected attribute. For this we extend the idea of null-space projections presented by Ravfogel et al. (2020) to the setting of kernel methods.

### 3.1. Motivation: Feature Space

At the heart of most kernel methods lies the kernel trick: many classical algorithms can ultimately be expressed solely in terms of inner products between data-points. Kernels exploit this by enabling efficient computation of inner products in potentially infinite-dimensional spaces. Formally, they are equivalent to performing an implicit pre-processing step in a (possibly infinite-dimensional) feature space. A kernel $k : \mathcal{X} \times \mathcal{X} \to \mathbb{R}$ is a map that factors through some inner product, i.e.

$$k(\mathbf{x}, \mathbf{x}') = \langle \Phi(\mathbf{x}), \Phi(\mathbf{x}') \rangle \tag{1}$$

where $\Phi$ is the feature map to some Hilbert space (the corresponding Hilbert space is referred to as feature space, the original space as input space). The most well known kernel is the rbf-kernel given by

$$k_{rbf}(\mathbf{x}, \mathbf{x}') \coloneqq \exp(-\gamma \|\mathbf{x} - \mathbf{x}'\|^2) \tag{2}$$

with parameter $\gamma$. This is also the one considered in this work and it does correspond to an infinite-dimensional feature space (Shawe-Taylor, 2004).

The naive extension of null-space projections to the respective feature spaces is only applicable if the dimension is finite and only tractable if quite small – as it is impossible to directly work with data living in an infinite dimensional space. However, there are two distinct ways this issue can be tackled: we usually only require the inner product between two projected points which can be implicitly expressed as a kernel recurrence or by using the so-called *empirical feature space* which is the approach taken in the following.

### 3.2. Fairness in the Empirical Feature Space

Given $n$ samples let $\mathbf{K} \in \mathbb{R}^{n \times n}$ denote the kernel matrix induced by a kernel $\mathbf{K}_{ij} = k(\mathbf{x}_i, \mathbf{x}_j)$ applied to the $d$-dimensional data stored in $\mathbf{X} \in \mathbb{R}^{n \times d}$. Our goal is to remove information about the protected attribute from $\mathbf{K}$ whilst keeping its required properties. As $\mathbf{K}$ is symmetric

positive-semidefinite it decomposes as

$$\mathbf{K} = \mathbf{Q}\boldsymbol{\Lambda}\mathbf{Q}^\top = \mathbf{G}\mathbf{G}^\top \qquad (3)$$

using the eigendecomposition of $\mathbf{K}$ and defining $\mathbf{G} := \mathbf{Q}\boldsymbol{\Lambda}^{1/2}$ where $\boldsymbol{\Lambda}^{1/2} := \mathrm{diag}(\sqrt{\lambda_1}, ..., \sqrt{\lambda_n})$. Note that the columns of $\mathbf{Q}$ correspond to the eigenvectors of $\mathbf{K}$. We view $\mathbf{G} = (\mathbf{g}_1, ..., \mathbf{g}_n)^\top \in \mathbb{R}^{n \times n}$ as an abstract data representation for the linear kernel, i.e., $k(\mathbf{x}_i, \mathbf{x}_j) = \langle \mathbf{g}_i, \mathbf{g}_j \rangle$ and $\mathbf{g}_i$ being a column vector. This means that we can write $\mathbf{G}\mathbf{G}^\top = \mathbf{K}$. This is called the empirical feature space which provides an alternative view on the kernel trick and is defined as $\mathbf{Y} := \mathbf{K}\mathbf{Q}\boldsymbol{\Lambda}^{-1/2}$ (Schölkopf et al., 1999; Xiong et al., 2005; Kwak, 2013). The connection to Equation (3) can be seen by observing that

$$\mathbf{K}\mathbf{Q}\boldsymbol{\Lambda}^{-1/2} = \mathbf{Q}\boldsymbol{\Lambda}\mathbf{Q}^\top\mathbf{Q}\boldsymbol{\Lambda}^{-1/2} = \mathbf{Q}\boldsymbol{\Lambda}^{1/2} = \mathbf{G}. \qquad (4)$$

This *empirical feature space* retains the geometric structures of the finite subspace spanned by the training data within the feature space (it is isometric) and corresponds to the subspace that a kernel method relies on for training (Schölkopf et al., 1999). As $\mathbf{g}_1, ..., \mathbf{g}_n$ is a finite dimensional representation we can directly apply subspace projections and apply the linear kernel afterwards. This is now done by null-space projections as presented by Ravfogel et al. (2020) for the case of regression. That is first we find a weighting that contains information for predicting the protected attribute $\mathbf{p}$

$$\min_{\mathbf{w}} \|\mathbf{p} - \mathbf{G}\mathbf{w}\|^2 + \tilde{\alpha}\|\mathbf{w}\|^2 \qquad (5)$$

by a default ridge-regression approach. The solution to this is straight-forward and well known as

$$\mathbf{w} = \mathbf{G}^\top(\mathbf{G}\mathbf{G}^\top + \tilde{\alpha}\mathrm{Id})^{-1}\mathbf{p}. \qquad (6)$$

Note that there are two equivalent ways of writing Equation (6), this version is required for the kernel trick. Similarly, we require this so the projection can later be expressed in terms of $\mathbf{K}$. Then we remove the information from $\mathbf{G}$ by enforcing

$$\mathbf{G}\mathbf{w} \overset{!}{=} \mathbf{0} \qquad (7)$$

with the null-space projection $\mathbf{P}^\mathbf{G}$ as follows:

$$\mathbf{G}_{(1)} = \mathbf{G}_{(0)}\mathbf{P}^\mathbf{G}_{(0)} \qquad (8)$$

where we define $\mathbf{P}^\mathbf{G}_{(1)} := (\mathrm{Id} - \mathbf{w}(\mathbf{w}^\top\mathbf{w})^{-1}\mathbf{w}^\top)$ as this enforces Equation (7) and where the scalar $(\mathbf{w}^\top\mathbf{w})^{-1}$ ensures normalization. Of course $\mathbf{G}_{(1)}$ might still contain information about the protected attribute so this process can be iterated by finding a new $\mathbf{w}$ and chaining the projection matrices:

$$\mathbf{G}_{(m)} = \mathbf{G}_{(0)}\prod_{i=0}^{m-1}\mathbf{P}^{\mathbf{G}_{(i)}}. \qquad (9)$$

The projected kernel matrix at the $m$-th iteration is then retrieved by

$$\mathbf{K}_{(m)} = \mathbf{G}_{(m)}\mathbf{G}_{(m)}^\top. \qquad (10)$$

which can now be used as a kernel for training a prediction task with respect to the target. Note that $\mathbf{K}_{(0)}$ corresponds to the original kernel matrix. Yet, it is not clear whether the product $\prod_{i=0}^{m-1}\mathbf{P}^{\mathbf{G}_{(i)}}$ is a valid projection matrix as in general the product of projections does not need to be a projection itself. This is ensured by the subsequent lemma following the approach taken in Ravfogel et al. (2020).

**Lemma 3.1.** $\prod_{i=0}^{m-1}\mathbf{P}^{\mathbf{G}_{(i)}}$ *is a projection.*

*Proof.* All proofs are found in Appendix A. $\square$

To perform the projection for $k$ unseen test points they have to be first mapped to the empirical feature space (see Xiong et al. (2005); Kwak (2013)), then the projection in Equation (9) performed and the linear kernel applied. This detour can be avoided by expressing this in a single transformation directly in terms of the kernel matrix which is one of our key contributions and is given by the following theorem:

**Theorem 3.2.** *Let* $\mathbf{K}_{(m)}$ *be defined as in Equation* (10). *Then for* $m > 0$ *it holds:*

$$\mathbf{K}_{(m)} = \mathbf{K}_{(m-1)}\mathbf{T}^{\mathbf{K}_{(m-1)}} \qquad (11)$$

*where we set* $\tau_{norm} := (\mathbf{w}^T\mathbf{w})^{-1}$ *and define*

$$\mathbf{T}^{\mathbf{K}_{(m)}} := (\mathrm{Id} - \mathbf{M}\mathbf{K}_{(m)})$$

$$\mathbf{M} := (\mathbf{K}_{(m)} + \tilde{\alpha}\mathrm{Id})^{-1}\mathbf{p}\tau_{norm}\mathbf{p}^\top(\mathbf{K}_{(m)} + \tilde{\alpha}\mathrm{Id})^{-1}.$$

Algorithm 1 provides pseudo-code of this procedure. The following corollary emphasizes that the resulting kernel matrix *firstly* still corresponds to the inner product in some feature space and thus *secondly* ensures necessary properties such as convexity in subsequent algorithms (such as SVR).

**Corollary 3.3.** $\mathbf{K}_{(m)}$ *in eq.* (11) *is positive semi-definite.*

Note that this also extends to the case of multiple protected attributes (see Appendix A.2). Theorem 3.2 yields a transformation $\mathbf{T}^{\mathbf{K}_{(m-1)}}$ for a single iteration which removes the relevant information on per-row basis whilst maintaining positive semi-definiteness[3]. Most importantly this can be directly applied to $\mathbf{K}_{test} \in \mathbb{R}^{k \times n}$. The transformations can be combined to yield a single transformation w.r.t. $\mathbf{K}_{(0)}$ as

$$\mathbf{T}_m := \prod_{i=0}^{m-1}\mathbf{T}^{\mathbf{K}_{(i)}}. \qquad (12)$$

---

[3]A kernel $k$ is usually called valid or positive definite if the corresponding kernel matrix is positive semi-definite for all finite number of points.

---

**Algorithm 1** Fair Kernel Decomposition

---

1: **Input:** $n \times n$ Kernel matrix $\mathbf{K}$, $n \times l$ protected attribute(s) $\mathbf{p}$, number of iterations $m$, regularization parameter $\tilde{\alpha}$.
2: $\mathbf{K}_0 \leftarrow \mathbf{K}; \mathbf{T}^{\mathbf{K}_0}, \mathbf{T}_0 \leftarrow \mathrm{Id}$
3: **for** $i \leftarrow 1$ to $m$ **do**
4: $\quad \mathbf{B} \leftarrow (\mathbf{K}_{(i-1)} + \tilde{\alpha}\mathrm{Id})^{-1}$ {Nystroem possible}
5: $\quad \tau_{norm} \leftarrow (\mathbf{p}^\top \mathbf{B}\mathbf{K}_{(i-1)}\mathbf{B}\mathbf{p})^{-1}$
6: $\quad \mathbf{M} \leftarrow \mathbf{B}\mathbf{p}\tau_{norm}\mathbf{p}^\top \mathbf{B}$
7: $\quad \mathbf{T}^{\mathbf{K}_i} \leftarrow \mathrm{Id} - \mathbf{M}\mathbf{K}_{(i-1)}$ {Trf. from $\mathbf{K}_{(i-1)}$ to $\mathbf{K}_{(i)}$}
8: $\quad \mathbf{K}_{(i)} \leftarrow \mathbf{K}_{(i-1)}\mathbf{T}^{\mathbf{K}_{(i)}}$
9: $\quad \mathbf{T}_{(i)} \leftarrow \mathbf{T}_{(i-1)}\mathbf{T}^{\mathbf{K}_{(i)}}$ {Trf. from $\mathbf{K}_{(0)}$ to $\mathbf{K}_{(i)}$}
10: **end for**
11: **Output:** Transformed positive semi-definite kernel matrix $\mathbf{K}_{(m)}$, transformation matrix $\mathbf{T}_{(m)}$.

---

so that it holds $\mathbf{K}_{(m)} = \mathbf{K}_{(0)}\mathbf{T}_m$. When we refer to iteration $m$ of our approach we mean the transformation $\mathbf{T}_m$. Note that $\mathbf{T}_m$ corresponds to the linear kernel applied to the iterated projection of the empirical feature space – $\mathbf{T}_m$ itself is not a projection. Alternatively it is possible to look at this directly in the feature space but this is beyond the scope of this work. Note that in contrast to Pérez-Suay et al. (2017) we directly find a predictive direction as weights of a model and Tan et al. (2020) instead aim to align different subspaces of the hypothesis space.

### 3.2.1. CHOICE OF ALGORITHM

Even though we rely on a ridge-regression procedure for the null-space projection this method only directly affects the kernel so it can be plugged into any subsequent algorithm relying on a kernel matrix. In this work we focus on the two canonical classical approaches: Kernel Ridge Regression (KRR) and Support Vector Regression (SVR).

### 3.2.2. COMPUTATIONAL COMPLEXITY AND NYSTROEM

In terms of complexity, our approach requires to invert the $n \times n$ kernel matrix (usually $\mathcal{O}(n^3)$). This expensive operation can be avoided using the Nystroem approximation (Drineas & Mahoney, 2005) for the inverse $(\mathbf{K}_{(m-1)} + \tilde{\alpha}\mathrm{Id})^{-1}$ in Theorem 3.2. Further, computational complexity from matrix multiplications can be reduced by choosing an appropriate order for multiplication and by not storing the transformations $\mathbf{T}_{(i)}$ explicitly – we provide the full details in Appendix C due to space restrictions.

We evaluate the Nystroem inverse approximation in Section 4.5. However, a remaining limitation is that the kernel matrix is still explicitly stored in memory which quickly becomes intractable for large datasets as the kernel matrix requires $\mathcal{O}(n^2)$ storage – this limitation is inherent to kernel methods but could be improved by aiming to directly ap-

proximate the kernel matrix instead of only its inverse. This direction for improvement is however left for future work.

## 4. Experimental Results

All implementations as well as all experiments in this section are made publicly available on GitHub.[4]

### 4.1. Real-world Datasets

We consider the following datasets:

- *Communities & Crimes:* This dataset is provided by Redmond & Baveja (2002) and widely used in the continuous fairness setup. The premise is to predict the "crime rate" based on characteristics of a given community. The protected attribute is the "percentage of black people".

- *ACSIncome:* Ding et al. (2021) introduce novel datasets based on data from the "American Community Survey" and is distinguished by US state. The "ACSIncome" task aims to predict the income of a person based on education, work hours and other social characteristics. We consider "age" as the protected attribute. The size of the data varies considerably between different states, to render computations tractable we choose the smaller dataset corresponding to "Montana".

- *ACSTravelTime:* This is also introduced by Ding et al. (2021) with the aim of predicting the "time to commute" based on social and personal characteristics. We consider "age" as protected attribute and also choose the state "Montana".

Unfortunately there is a lack of datasets for fairness in general and it is up for debate how sensible (see for example Bao et al. (2021)) existing and widely used datasets are – an issue exacerbated by the "continuous fairness" setting. However, the ACS data is commonly used in social fields (refer to Section 2.3). Each problem discussed above aims at possible use-cases in the real-world: city planners develop efficient traffic routing ("ACSTravelTime"), employers predict the salary of a new hire ("ACSIncome") and the police allocate its forces to certain regions ("Crimes"). In all these cases laws govern that there should be no discrimination based on "age" and "race" present as continuous attributes.

### 4.1.1. METHODS

We compare our approach in combination with KRR and SVR which we term "KRR/SVR-FKD" where "FKD" stands for "Fair Kernel Decomposition". To benchmark

---

[4]https://github.com/Felix-St/FairKernelDecomposition

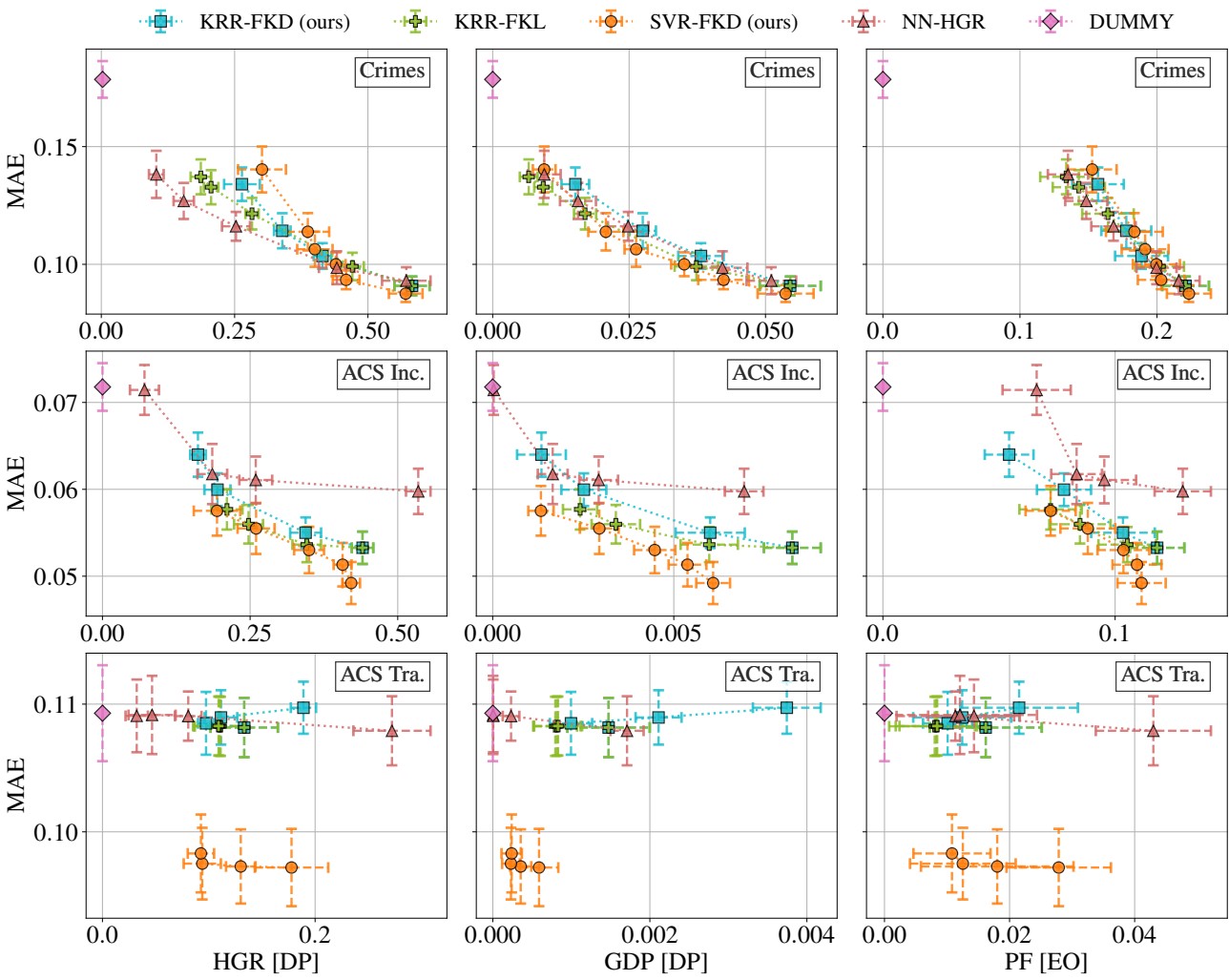

*Figure 1.* Pareto fronts for the MAE (y-axis, lower is better) and three different fairness measures (x-axis, lower is better) reported with empirical standard deviations (error-bars). Comparison for multiple methods (marker, color) and datasets (boxes, one per row). Number of iterations $m$ for *-FKD, regularization $\mu$ for *-FKL and HGR penalty $\lambda$ are found in Appendix B.

against another fairness approach for KRR we compare against Pérez-Suay et al. (2017) where we denote their approach as "KRR-FKL". Additionally we benchmark against the HGR penalized neural network from Mary et al. (2019) (which we denote "NN-HGR") as well as a dummy classifier that simply predicts the training's targets mean.

We do not compare against Perez-Suay et al. (2023) as they require label information for test data and found the most recent approach (Kong et al., 2025) to not outperform the other baselines (see Appendix D for a comparison).

### 4.1.2. PARAMETERS

For the kernel methods we use the rbf-kernel (see Equation (2)). For every dataset we first run a grid-search on the non-fair base methods (KRR, SVR) to find their opti-

mal default hyper-parameters. For our approach we have an additional parameter $\tilde{\alpha}$ for the KRR used inside our decomposition approach which we set heuristically (for an analysis see Section 4.3). Otherwise we run our evaluation for a range of different regularization parameters (penalty coefficient and number of iterations). Details about the used parameters and more are given in Appendix B. For "NN-HGR" we choose the architecture reported in the original contribution (Mary et al., 2019).[5]

---

[5]Note that this architecture might not be optimal for every dataset but a fair comparison of completely different approaches is not straight-forward for multi-objective problems due to the complexity of hyperparameter optimization. Appendix D contains results with a larger network which does not improve performance.

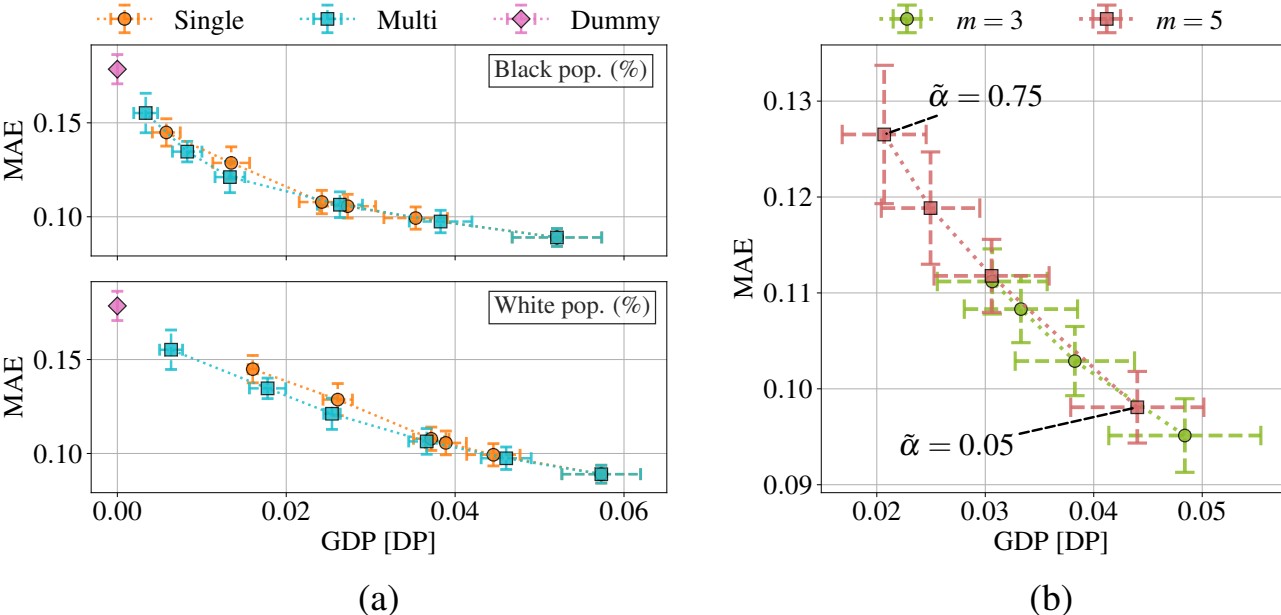

*Figure 2.* Other experiments on the "Crimes" dataset. (a): Performance (MAE, y-axis, lower is better) of our "SVR-FKD" considering both the white and black population percentage (pop. perc.) as protected (multi, squares, cyan, $m \in \{0, 5, 12, 25, 35, 45\}$) and only the black pop. perc. (single, circles, orange, $m \in \{0, 10, 35, 50, 70, 85\}$). GDP fairness (x-axis, lower is better) with respect to the black pop. perc. (top) and to the white pop. perc. (bottom) (b): Influence of $\tilde{\alpha}$ on the performance (MAE, y-axis, lower is better) of our "KRR-FKD" for fixed parameters and iterations $m \in \{3, 5\}$ (color and marker). GDP fairness (x-axis, lower is better) w.r.t the black pop. perc.

### 4.1.3. MEASURES

All reported results are obtained by a 5-fold cross valida-tion. We report the mean absolute error (MAE) for target evaluation and use the following measures for fairness (see Section 2.1 for the corresponding explanation and refer-ences): to evaluate for demographic parity we choose HGR (denoted HGR [DP]) and GDP (denoted GDP [DP]) whereas for equalized odds[6] we use pairwise fairness (denoted PF [EO]). The empirical standard deviation across all runs is provided for every measure and shown as error-bars.

### 4.1.4. RESULTS

Evaluations concerning fair machine learning require con-sidering a trade-off between predictive performance and fairness – hence we are interested in the corresponding (ap-proximate) pareto fronts which are visualized in Figure 1. Note that each fairness measure is taken at the same time for the same model i.e. every row evaluates the same models and same predictions but with different fairness measures.

On the *Crimes* dataset differences can be observed for the HGR measure where "NN-HGR" and "KRR-FKL" outper-form our approach in case of strong regularization despite that this margin is interestingly much closer for the other measures and our "SVR-FKD" outperforms the other ap-

---
[6]For pairwise fairness this is identical to equal opportunity.

proaches slightly for weak regularization being more pro-nounced for the GDP score.

On the *ACSIncome* dataset our "SVR-FKD" generally out-performs all other approaches particularly pronounced for GDP, whereas our "KRR-FKD" is outperformed by "KRR-FKL" both in turn providing better trade-offs than "NN-HGR" even for the HGR score.

On the *ACSTravelTime* dataset we observe that most meth-ods – even if unregularized – perform equally to the dummy regressor. This is not the case for "SVR-FKD" that also achieves considerable improvements in terms of fairness.

In conclusion, we can see quite heterogeneous results – em-phasizing that to achieve fairness the right method has to be chosen for the right task-at-hand. Our newly introduced methods show competitive performance across multiple measures and datasets with best performance for "SVR-FKD" thus providing an useful new method for fair ML.

### 4.2. Multiple Protected Attributes

Next to predictive performance we so-far optimized for fairness with respect to a single attribute. Of course it is possible we either fail to ensure fairness with respect to another attribute or maybe even introduce unwanted biases by discriminating against other groups.

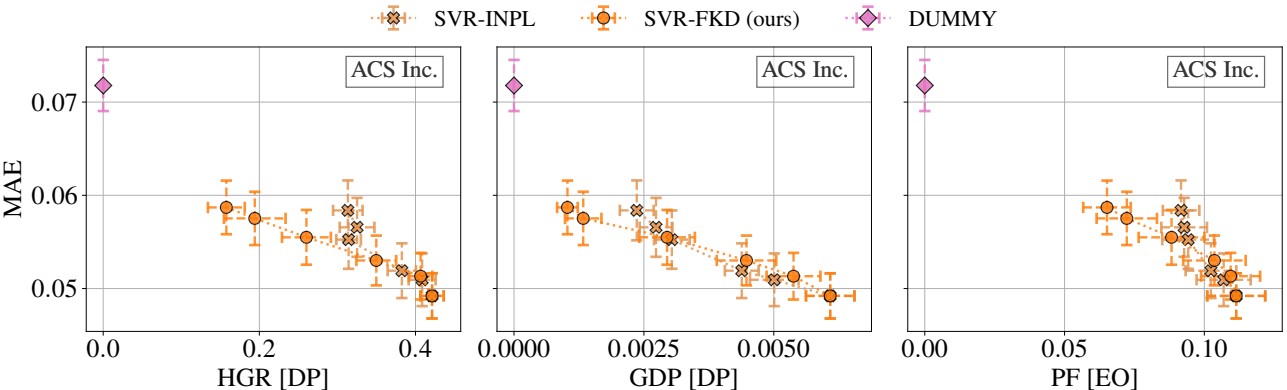

*Figure 3.* Same experimental setting as Figure 1. Comparison between SVR-FKD and linear-only removal (SVR-INPL).

### 4.2.1. EXPERIMENTAL SETUP

To analyze how our approach compares when considering multiple protected attributes we again utilize the "Crimes" dataset. Next to the "black population percentage" (pop. perc.) it also includes the "white population percentage". We compare the performance of our "SVR-FKD" once being trained to protect both attributes ("multi") and once only for the "black population percentage" ("single") and then measuring fairness with respect to each attribute. Otherwise the same experimental setup is used as previously.

### 4.2.2. RESULT

Figure 2 (a) shows how in both cases fairness improves for both groups which is sensible as there is a clear connection between the attributes. For the black pop. perc. we observe very similar fairness-scores but with slight differences for high regularization – this effect is more pronounced for the white pop. perc. where we do observe more noticeable differences as the multi-protected approach achieves improved trade-offs – even though the margin remains slim. The multi-protected setting is of interest for further investigation with wider availability of appropriate datasets in the future.

### 4.3. Influence of $\tilde{\alpha}$

A key parameter in our approach is the regularization $\tilde{\alpha}$ for the ridge-regression that computes the weight used for the null-space projection. Usually $\tilde{\alpha}$ controls for overfitting, however we might benefit from only removing very specific information about the protected attribute – rendering the interpretation more difficult.

We analyze this by measuring the performance of "KRR-FKD" for a fixed number of iterations on the "Crimes" dataset whilst varying $\tilde{\alpha}$ – this is shown in Figure 2 (b). The parameter plays a crucial role: the higher $\tilde{\alpha}$ the larger the change to predictive performance i.e. smaller values correspond to a more fine-grained removal of information.

This suggests using a smaller value and control the degree of regularization by choosing an appropriate $m$ – the precise choice, however, as always remains dataset specific.

### 4.4. Comparison to Linear Removal

Evidently, if the relationship between the data and the protected attribute is linear it is fully sufficient to rely on linear removal. This poses the question of the advantage provided by our approach if this relationship is indeed non-linear which we investigate in the following.

### 4.4.1. EXPERIMENTAL SETUP

This is done on *ACSIncome* by comparing our "SVR-FKD" to a linear baseline ("SVR-INPL") which implements the same idea of null-space projections, up to specific modifications to improve numerical stability. Crucially the projection is applied directly to the data — the projection is thus linear — then a (non-linear) Kernel SVR is applied to the projected data. This differs from using "SVR-FKD" in conjunction with a linear kernel as this happens in the same kernel space; thus using a linear kernel in our projection would also force us to use a linear kernel in the predictive model.

Both approaches use the same parameters as listed in Appendix B with the exception that we use some additional iterations, i.e. $m \in \{0, 60, 100, 160, 180, 200\}$ for "SVR-INPL" and $m \in \{0, 45, 60, 80, 140\}$ for "SVR-FKD".

### 4.4.2. RESULT

Figure 3 highlights how at the beginning "SVR-INPL" is able to remove the linear dependencies even though the improvements in fairness for the same $m$ are much smaller than for "SVR-FKD", indicating that the amount of information removed about the protected attribute is smaller at each step for "SVR-INPL". At a certain point, the MAE keeps increasing without improvements in fairness for "SVR-INPL" (even with a larger number of iterations e.g., 160, 180, 200)

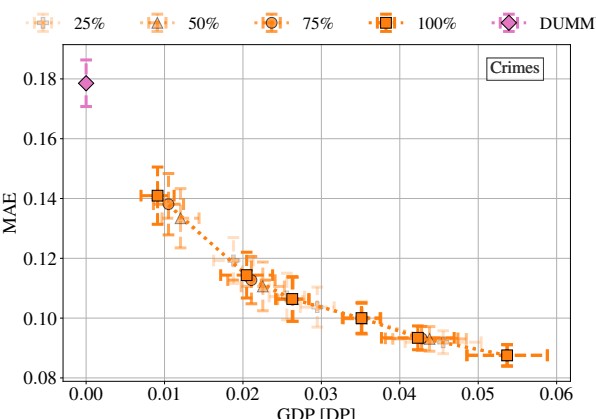

*Figure 4.* Nystroem approximation with different percentages (denoted by transparency and marker) of components for "SVR-FKD". "Crimes" with the same experimental setting as before.

whereas our approach, despite also increasing in MAE, keeps improving in fairness as the linear approach fails to capture the non-linear dependencies. This effect is significantly more pronounced for HGR and PF than for GDP.

### 4.5. Nystroem Inverse Approximation

To address the scalability issues we test the performance difference if we rely on the Nystroem inverse approximation. We use the same settings as before, focus on the "Crimes" dataset and show the performance for different levels of approximation by using different percentages of the original number of components for the approximation.

Figure 4 clearly demonstrates qualitatively close results even if some differences persist, the overall trend is identical. It is of great interest to investigate the limits of the scalability in future – currently Nystroem is used for the inverse of the kernel matrix, a possible improvement would be to perform the projections directly in the reduced representation or by using completely different approaches for the kernel approximation.

## 5. Conclusion

This work introduces an approach that allows the extension of null-space projections for fairness to the more general case of kernel methods for regression with continuous protected attributes. We term this setting "continuous fairness" which is often neglected within the ML community (Section 2.3). Our approach is model and task agnostic as it directly depends on the kernel matrix allowing for a natural extension to unseen data.

We empirically demonstrate that our approach is suited for the application of different regression strategies using the transformed kernel matrix. This yields competitive perfor-

mance and, in the case of our newly introduced approach for fair Support Vector Regression with continuous protected attributes ("SVR-FKD"), improved performance.

### 5.1. Limitations and Future Work

Some limitations are inherent to our approach, such as the iterative and kernel based nature: for $m$ iterations and $n$ samples we require $\mathcal{O}(m \cdot n^3)$ operations. We empirically found that this can be significantly reduced by the Nystroem method for matrix inversion without a significant loss in performance: however we still require $\mathcal{O}(n^2)$ in storage thus limiting scalability. On this end, it is of interest to analyze the possibility of going beyond approximating the inverse of the kernel matrix by directly performing the null-space projections in the reduced Nystroem representation or by using a different form of kernel approximation altogether (such as random Fourier features (Rahimi & Recht, 2007)).

Future work might focus on analyzing the performance on other *kernel methods* we did not investigate in this work such as Gaussian Processes.

Another interesting extension and modification of our approach would be to other *tasks* such as classification with a continuous protected attribute, regression with a discrete protected attribute or in other domains such as privacy.

A future analysis of the alternative viewpoint of direct projection in the feature space is another direction of interest.

## Acknowledgements

Funding in the scope of the ERC Synergy Grant "Water-Futures" No. 951424 is gratefully acknowledged.

## Impact Statement

This paper presents work whose goal is to advance the field of (Algorithmic) Fair Machine Learning. This includes systems that provide automatic decisions with potential impact on humans – we do not argue in favor of, or against the use of such automated decision systems. But reality is that such ML systems are in use and that they – as do ML systems in general – already affect society in a variety of ways. Further, as with most fairness approaches it is possible to misuse the introduced algorithm – for example to wrongfully remove the contributions of marginalized groups.

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

## A. Proofs

### A.1. Proof for Lemma 3.1:

*Proof.* Following the proof in Ravfogel et al. (2020, Appendix A) we only need to show that we have $\mathbf{w}_1^\top \mathbf{w}_2 = 0$ for two consecutive iterations of our algorithm. After finding $\mathbf{w}_1$ with Equation (6) and projection in Equation (8) we obtain $\mathbf{G}_{(1)}$ such that $\mathbf{G}_{(1)}\mathbf{w}_1 = 0$ (∗). Iterating we now find $\mathbf{w}_2$ using $\mathbf{G}_{(1)}$. By the representer theorem (Schölkopf et al., 2001) this can be written as a finite linear combination $\mathbf{w}_2 = \sum_{i=1}^n \xi_i \mathbf{g}_i$ where the $\mathbf{g}_i$ correspond to the rows in $\mathbf{G}_{(1)}$. By (∗) it holds: $\mathbf{g}_i^\top \mathbf{w}_1 = \mathbf{w}_1^\top \mathbf{g}_i = 0$ and therefore:

$$\mathbf{w}_1^\top \mathbf{w}_2 = \mathbf{w}_1^\top \sum_{i=1}^n \xi_i \mathbf{g}_i = \sum_{i=1}^n \xi_i \mathbf{w}_1^\top \mathbf{g}_i = 0.$$

The full result follows by the same argument and corollaries that are found in Ravfogel et al. (2020, A.1.1-A.1.3). □

### A.2. Proof for Theorem 3.2:

*Proof.* Let $\mathbf{P}_* := (\mathrm{Id} - \mathbf{w}(\mathbf{w}^\top \mathbf{w})^{-1}\mathbf{w}^\top)$, then $\mathbf{P}_*$ is a projection matrix so it holds:

$$
\begin{aligned}
\mathbf{K}_{(m)} &= \mathbf{G}_{(m-1)}\mathbf{G}_{(m-1)}^\top \\
&= \mathbf{G}_{(m-1)}\mathbf{P}_*(\mathbf{G}_{(m-1)}\mathbf{P}_*)^\top \\
&= \mathbf{G}_{(m-1)}\mathbf{P}_*\mathbf{P}_*^\top \mathbf{G}_{(m-1)}^\top \\
&= \mathbf{G}_{(m-1)}\mathbf{P}_*\mathbf{G}_{(m-1)}^\top
\end{aligned}
$$

where we used the properties of a projection matrix $\mathbf{P}_*^\top = \mathbf{P}_*$ and $\mathbf{P}_*^2 = \mathbf{P}_*$. Now simplifying $\mathbf{P}_*$ we can rewrite (for easier notation we omit the index $m-1$ here):

$$
\begin{aligned}
\mathbf{w}\tau_{norm}\mathbf{w}^\top &= \mathbf{G}^\top(\mathbf{G}\mathbf{G}^\top + \tilde{\alpha}\mathrm{Id})^{-1}\mathbf{p}\tau_{norm}(\mathbf{G}^\top(\mathbf{G}\mathbf{G}^\top + \tilde{\alpha}\mathrm{Id})^{-1}\mathbf{p})^\top \\
&= \mathbf{G}^\top(\mathbf{G}\mathbf{G}^\top + \tilde{\alpha}\mathrm{Id})^{-1}\mathbf{p}\tau_{norm}\mathbf{p}^\top(\mathbf{G}\mathbf{G}^\top + \tilde{\alpha}\mathrm{Id})^{-1}\mathbf{G} \\
&= \mathbf{G}^\top(\mathbf{K} + \tilde{\alpha}\mathrm{Id})^{-1}\mathbf{p}\tau_{norm}\mathbf{p}^\top(\mathbf{K} + \tilde{\alpha}\mathrm{Id})^{-1}\mathbf{G} \\
&= \mathbf{G}^\top\mathbf{M}\mathbf{G}
\end{aligned}
$$

where we define $\mathbf{M} := (\mathbf{K} + \tilde{\alpha}\mathrm{Id})^{-1}\mathbf{p}\tau_{norm}\mathbf{p}^\top(\mathbf{K} + \tilde{\alpha}\mathrm{Id})^{-1}$ and denote the normalization scalar by $\tau_{norm}$ and write it as

$$\tau_{norm} = (\mathbf{w}^\top \mathbf{w})^{-1} = \mathbf{p}^\top(\mathbf{K} + \tilde{\alpha}\mathrm{Id})^{-1}\mathbf{K}(\mathbf{K} + \tilde{\alpha}\mathrm{Id})^{-1}\mathbf{p}.$$

Further we have (again omitting the index):

$$\mathbf{G}\mathbf{P}_*\mathbf{G}^\top = \mathbf{G}(\mathrm{Id} - \mathbf{G}^\top\mathbf{M}\mathbf{G})\mathbf{G}^\top = (\mathbf{G} - \mathbf{G}\mathbf{G}^\top\mathbf{M}\mathbf{G})\mathbf{G}^\top = (\mathbf{G}\mathbf{G}^\top - \mathbf{G}\mathbf{G}^\top\mathbf{M}\mathbf{G}\mathbf{G}^\top) =: (*)$$

hence with $\mathbf{K}_{(m-1)} = \mathbf{G}_{(m-1)}\mathbf{G}_{(m-1)}^\top$ we have

$$(*) = \mathbf{K}_{(m-1)} - \mathbf{K}_{(m-1)}\mathbf{M}\mathbf{K}_{(m-1)} = \mathbf{K}_{(m-1)}(\mathrm{Id} - \mathbf{M}\mathbf{K}_{(m-1)})$$

which was to be shown. □

This setup can be naturally extended to the case of multiple protected attributes by instead letting $\mathbf{W} \in \mathbb{R}^{n \times l}$ aiming to predict $\mathbf{p} \in \mathbb{R}^{n \times l}$. Then instead of the scalar $\tau_{norm}$ for the normalization we have a matrix $\mathbb{R}^{l \times l} \ni \mathcal{T} = (\mathbf{W}^\top \mathbf{W})^{-1}$ which we assume to be invertible (this is usually the case but not, for example, if the protected attributes are linearly dependent). Therefore Theorem 3.2 naturally transfers.

### A.3. Proof for Corollary 3.3:

*Proof.* By Equation (10) and Theorem 3.2 $\mathbf{K}_{(m)}$ is equivalent to the linear kernel applied to $\mathbf{G}_{(m)}$. □

| Alg. | Dataset | Parameters |
|------|---------|------------|
| KRR-FKD (ours) | All | $\alpha = 0.25, \gamma = 0.05, \tilde{\alpha} = 0.1$ |
| KRR-FKL | All | Equal to KRR-FKD, $\tilde{\alpha}$ not used. |
| SVR-FKD (ours) | Crimes ACSInc ACSTra All | $\epsilon = 0.01, \gamma = 0.05, C = 0.75$ $\epsilon = 0.005, \gamma = 0.05, C = 0.5$ $\epsilon = 0.001, \gamma = 0.01, C = 0.125$ $\tilde{\alpha} = 0.05$ |
| NN-HGR | All | Adam, $\rho = 0.01, \eta = 1e-3$, SELU, Top.: $n \times 100 \times 80 \times 1$ Training for 500 epochs. |

*Table A.1.* Fixed hyper-parameters for the considered approaches. RBF parameter $\gamma$, ridge penalty $\alpha$, ridge penalty for fairness $\tilde{\alpha}$, epsilon bandwidth $\epsilon$, learning rate $\eta$ and weight decay $\rho$. $n$ inputs. Adam optimizer and SELU activation.

| Alg. | Dataset | Regularization |
|------|---------|----------------|
| KRR-FKD (ours) | Crimes ACSInc ACSTra | $m \in \{0, 5, 10, 18\}$ $m \in \{0, 15, 20, 30\}$ $m \in \{0, 15, 20, 30\}$ |
| KRR-FKL | Crimes ACSInc ACSTra | $\mu \in \{0, 0.05, 0.25, 0.6, 1\}$ $\mu \in \{0, 0.05, 0.25, 1\}$ $\mu \in \{0, 0.5, 1, 2\}$ |
| SVR-FKD (ours) | Crimes ACSInc ACSTra | $m \in \{0, 5, 30, 45, 60, 80\}$ $m \in \{0, 45, 60, 80, 100\}$ $m \in \{0, 45, 60, 100\}$ |
| NN-HGR | Crimes ACSInc ACSTra | $\lambda \in \{0, 0.025, 0.1, 0.2, 0.5\}$ $\lambda \in \{0, 0.00625, 0.0125, 0.05\}$ $\lambda \in \{0, 0.00625, 0.0125, 0.05\}$ |

*Table A.2.* Fairness regularization parameters that control the Fairness-Accuracy-Tradeoff. Number of iterations $m$ and penalty coefficients $\lambda, \mu$.

## B. Experimental Details

For the "NN-HGR" we use the implementation provided by the original paper (Mary et al., 2019). The Support Vector and Kernel Ridge Regression used for training our approach with the modified kernel utilize scikit-learn (Pedregosa et al., 2011). Table A.1 lists the fixed hyper-parameters that do not directly control the degree of fairness regularization. The different used fairness parameters to attain the results shown in Figure 1 are listed in Table A.2.

## C. Nystroem Approximation

The Nystroem approximation of a kernel matrix $\mathbf{K} \in \mathbb{R}^{n \times n}$ works by sampling a number of columns $p \ll n$ from $\mathbf{K}$. The indices of the columns are also referred to as "landmarks". With these we get the approximation (Williams & Seeger, 2000)

$$\mathbf{K} \approx \mathbf{K}_{np} \mathbf{K}_{pp}^{-1} \mathbf{K}_{pn}$$

which is exact if $p = n$ components are used or if $\mathbf{K}$ is of rank $p$ (and $p$ linearly independent columns were chosen). $\mathbf{K}_{pp}$ corresponds the intersection of the $p$ chosen rows and columns, $\mathbf{K}_{np}$ to the $p$ columns which are $n$ dimensional each. The matrix inversion lemma is given by (Rasmussen & Williams, 2008, p. 201)

$$(\mathbf{Z} + \mathbf{U}\mathbf{W}\mathbf{V}^\top)^{-1} =$$
$$\mathbf{Z}^{-1} - \mathbf{Z}^{-1}\mathbf{U}(\mathbf{W}^{-1} + \mathbf{V}^\top\mathbf{Z}^{-1}\mathbf{U})^{-1}\mathbf{V}^\top\mathbf{Z}^{-1}$$

which with the following definitions $\mathbf{Z} := \alpha\mathbf{I}$, $\mathbf{U} := \mathbf{K}_{np}$, $\mathbf{V} := \mathbf{K}_{np}$, $\mathbf{W} := \mathbf{K}_{pp}^{-1}$ yields:

$$(\alpha\text{Id} + \mathbf{K})^{-1} \approx (\alpha\text{Id} + \mathbf{K}_{np}\mathbf{K}_{pp}^{-1}\mathbf{K}_{pn})^{-1}$$
$$= (\mathbf{Z} + \mathbf{U}\mathbf{W}\mathbf{V}^\top)^{-1}$$
$$= \frac{1}{\alpha}\text{Id} - \frac{1}{\alpha^2}\mathbf{K}_{np}(\mathbf{K}_{pp} + \frac{1}{\alpha}\mathbf{K}_{pn}\mathbf{K}_{np})^{-1}\mathbf{K}_{pn}.$$

Evidently this only requires to invert a $p \times p$ matrix which is in $\mathcal{O}(p^3)$. Note that $\mathbf{K}_{pn} = \mathbf{K}_{np}^\top$. Of course this is an approximation so numerical issues might occur, to prevent some of them we first perform the approximation and then return the symmetrized result – see Algorithm 2.

Further, by choosing the optimal order of matrix multiplication we end up with an additional $\mathcal{O}(n^2(p + l))$ ($l$ protected attributes) in matrix multiplications if the transformations $\mathbf{T}_{(i)}$ are not stored explicitly. This is detailed in the code.

---

**Algorithm 2** Nystroem Approx. for Matrix Inversion

---

1: **Input:** Kernel matrix $\mathbf{K} \in \mathbb{R}^{n \times n}$, reg. param. $\alpha$
2: Draw $p$ column/landmark indices $c \leftarrow [c_1, ..., c_p]$
3: Set $\mathbf{K}_{np} \in \mathbb{R}^{n \times p}$ with entries $\mathbf{K}_{np}[i,j] \leftarrow \mathbf{K}[i, c_j]$
4: Set $\mathbf{K}_{pp} \in \mathbb{R}^{p \times p}$ with entries $\mathbf{K}_{pp}[i,j] \leftarrow \mathbf{K}[c_i, c_j]$
5: $\mathbf{A} \leftarrow (\mathbf{K}_{pp} + \frac{1}{\alpha}\mathbf{K}_{pn}\mathbf{K}_{np})^{-1}$ $\{p \times p$ inverse$\}$
6: $\mathbf{B} \leftarrow \frac{1}{\alpha}\mathrm{Id} - \frac{1}{\alpha^2}\mathbf{K}_{np}\mathbf{A}\mathbf{K}_{pn}$
7: $\mathbf{B} \leftarrow \frac{\mathbf{B}+\mathbf{B}^\top}{2}$ $\{$ensure symmetry for numerical stability$\}$
8: **Output:** $\mathbf{B}$ the approx. of the inverse $(\mathbf{K} + \alpha\mathrm{Id})^{-1}$

---

## D. Additional Experiments

In our experiments we also bench-marked against the most recent method (Kong et al., 2025) for continuous fairness which we denote by "NN-FREM" but found it to be surpassed by the other baselines ("KRR-FKL"), see Figure 5. We used the same fixed network parameters (Table A.1) as for "NN-HGR" and used the following method-specific parameters: $\gamma = 0.05$, $\sigma = 1$, $\lambda \in \{0, 0.5, 1.5, 5, 10\}$ across all datasets. We also include "NN-HGR-L" which is the same as "NN-HGR" but using a larger topology of size $n \times 250 \times 230 \times 1$ to demonstrate that increasing model complexity does not help.

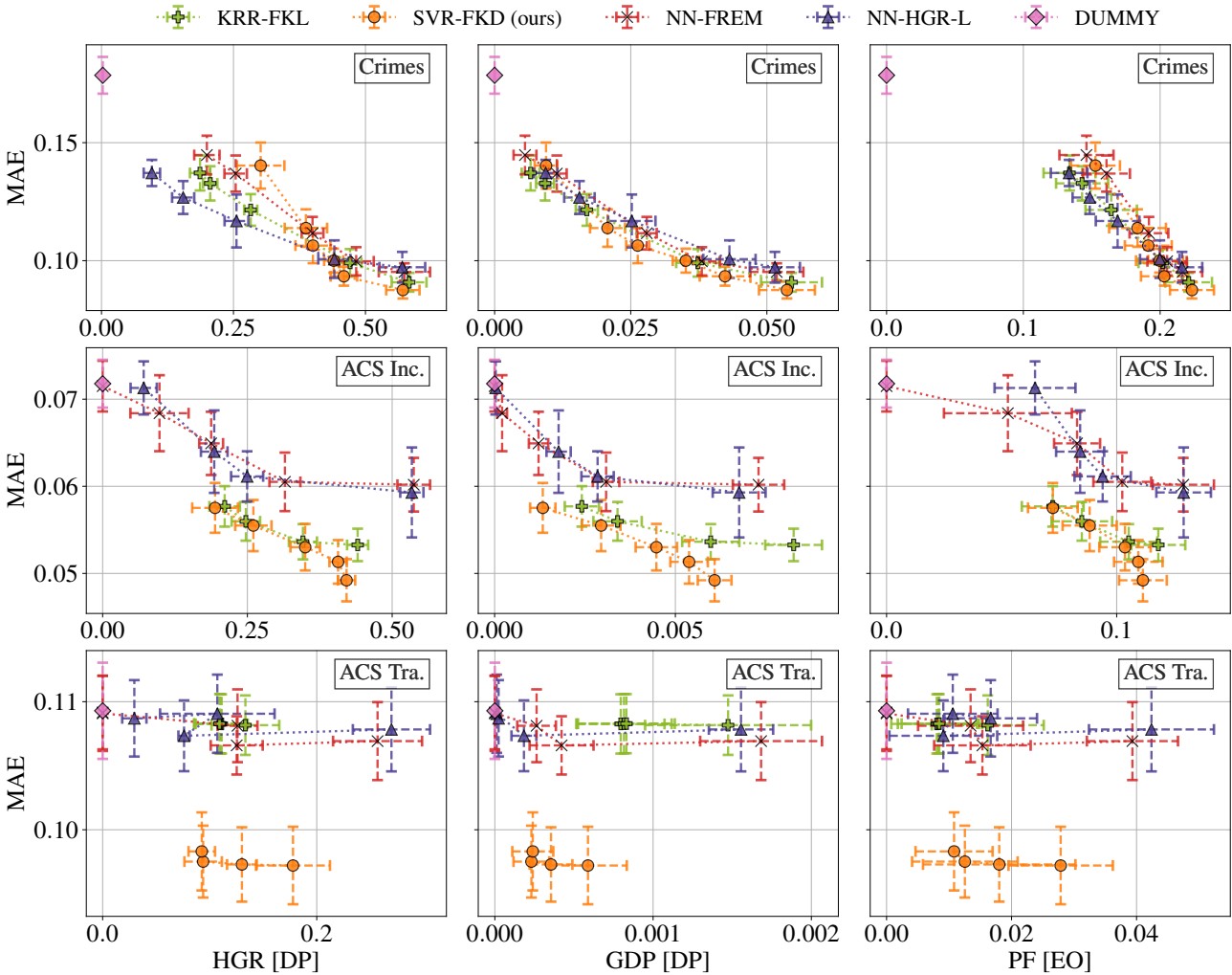

*Figure 5.* Same experimental setup as in Figure 1. Selected approaches compared to the most recent "NN-FREM" by Kong et al. (2025).

