# OpenReview forum: "Extending Fair Null-Space Projections for Continuous Attributes to Kernel Methods"
_ICML.cc/2026/Conference — ICML 2026 regular_

### Official Review · Reviewer_frFe · 2026-03-11

**Soundness:** 3
**Presentation:** 3
**Significance:** 3
**Originality:** 3
**Overall Recommendation:** 4
**Confidence:** 4

**Summary:**

This paper addresses the "continuous fairness" problem in machine learning regression tasks. While existing fairness literature primarily focuses on discrete attributes or linear models, this work extends iterative null-space projections to kernel methods via the "empirical feature space." The authors theoretically prove that this process is equivalent to a direct transformation of the kernel matrix, yielding a method for handling continuous protected attributes that is agnostic to both the specific predictive model and the fairness metric. Experimental results demonstrate that this framework (e.g., when combined with Support Vector Regression) achieves competitive or superior performance on multiple benchmark datasets compared to existing methods.

**Compliance With Llm Reviewing Policy:**

Affirmed.

**Key Questions For Authors:**

The formatting of several figures could be improved: Figures 1 and 2 suffer from overly large internal fonts due to improper scaling, and there is excessive whitespace between Figure 3 and the main text.

**Limitations:**

yes

**Strengths And Weaknesses:**

### strengths
1. This paper provides a theoretical derivation that translates null-space projection in the empirical feature space into an algebraic transformation of the kernel matrix, preserving properties such as positive semi-definiteness and enabling out-of-sample extension.
2. The proposed transformation exhibits strong model-agnosticism, providing a highly versatile and effective approach for constructing various fair kernel-based models (e.g., fair Support Vector Regression).
### weaknesses
1.  The debiasing process based on null-space projection lacks feature-level interpretability. Since the projections are performed in the empirical feature space, the resulting kernel matrix satisfies mathematical fairness, but it is difficult to identify which input features are effectively suppressed. This black-box mechanism may raise interpretability and regulatory concerns in applications requiring transparency, such as credit approval or judicial decision-making.
2.  Although scalability is claimed, the authors restricted the ACS datasets to a single state (Montana) to "render computations tractable". This highlights a significant $\mathcal{O}(n^3)$ computational bottleneck. The absence of full-scale experiments leaves the method's practical scalability unverified.
3.  The comparison with the deep learning baseline (NN-HGR) is potentially unfair. The authors reused a fixed network architecture ($n \times 100 \times 80 \times 1$) without dataset-specific tuning, explicitly acknowledging its potential sub-optimality in a footnote (Section 4.1). This lack of proper hyperparameter search likely degrades the baseline's performance and weakens the reliability of the comparative results.

---

> ### Author Rebuttal · Authors · 2026-03-29
>
> Thank you for your time taken to provide a thorough review, as well as for acknowledging the versatility of our approach!
>
> **Feature Interpretability**
>
> > The debiasing process based on null-space projection lacks feature-level interpretability.
>
> This is true. This corresponds to the usual tradeoff in performance vs. interpretability when replacing linear approaches by non-linear ones (see also our answer to reviewer sv4m for a comparison to a linear removal only, we will mention this reduction in interpretability in the final version).
>
> **HGR Architecture Additional Experiment**
>
> >  This lack of proper hyperparameter search likely degrades the baseline's performance and weakens the reliability of the comparative results.
>
> We tested larger architectures but did not find significant differences, the following anonymous link contains results for NN-HGR with a larger network architecture (denoted ``NN-HGR-L'') of size $n\times 250 \times 230 \times 1$: https://anonymous.4open.science/r/RebuttalICML-FF7C/Rebuttal_HGR-L.pdf.
>
>
> **On Scalability**
>
> > Although scalability is claimed, the authors restricted the ACS datasets to a single state (Montana) to "render computations tractable". This highlights a significant computational bottleneck. The absence of full-scale experiments leaves the method's practical scalability unverified.
>
>  We acknowledge the limitations of kernel based approaches as storing the kernel matrix naively already requires $O(n^2)$ storage thus limiting the applicability to larger datasets.
>  Yet, we want to stress that this limitation is inherent to (exact) kernel based approaches in general.
>
>  The run-time complexity is improved to $O(n^2)$ with similiar performance as illustrated in section 4.4 and Appendix C by the Nystroem approximation for the kernel matrix inverse. Further reduction, also in storage complexity, could be possible by adapting the projection to work directly with the reduced representation or by using other kernel approximation techniques (e.g. Random Fourier Features): this investigation is left for future work.
>
> We will expand the already existing discussion on these limitations as well as possible paths forward for improvements of the approximation technique in the final version of the paper.
>
> **Formatting**
>
> > The formatting of several figures could be improved: Figures 1 and 2 suffer from overly large internal fonts due to improper scaling, and there is excessive whitespace between Figure 3 and the main text.
>
> Thank you for the input, we will reduce the font size of the x and y labels of Figures 1 and 2 to better match the size of the text and increase the size of Figure 3 to reduce whitespace.

---

> > ### Author Rebuttal · Reviewer_frFe · 2026-04-03
> >
> > N/A

---

### Official Review · Reviewer_sv4m · 2026-03-12

**Soundness:** 3
**Presentation:** 3
**Significance:** 3
**Originality:** 3
**Overall Recommendation:** 5
**Confidence:** 3

**Summary:**

This work aims to enforce fairness in kernel-based models with continuous protected attributes by mitigating dependence between the representation and the protected variable. It iteratively removes directions in the empirical feature space that allow prediction of the protected attribute and then train a standard kernel algorithm on this transformed kernel to predict the task.  Experiments on three real-world datasets show competitive fairness–accuracy trade-offs compared to existing approaches.

**Compliance With Llm Reviewing Policy:**

Affirmed.

**Final Justification:**

The paper is technically solid, proposes a model-agnostic approach with competitive results, and is clearly presented. The rebuttal addressed my main question by adding the linear projection baseline comparison and helped clarify the advantage of the proposed method. I have therefore increased my score.

My remaining concern is that this method does not directly optimize the fairness-accuracy trade-off and may discard information that is useful for predicting Y.

**Key Questions For Authors:**

1)  How does the proposed kernelized approach compare empirically to a linear null-space projection baseline applied directly in the original feature space? Does using a kernel representation provide a clear advantage in reducing dependence on p while preserving predictive performance?

2) Given the connection to Kernelized Concept Erasure (Ravfogel et al., 2022), could the authors provide an empirical comparison with that method? In particular, how do they compare in terms of the fairness–accuracy trade-off?

**Limitations:**

1) The method does not guarantee independence from the protected attribute, as it only removes directions that are linearly predictive of p in the RKHS,so nonlinear dependencies in this representation may still remain in the final predictions.
2) The method removes information about the protected attribute p without considering the prediction target Y, meaning it does not explicitly optimize the fairness–accuracy tradeoff and may discard information that is useful for predicting Y.
3) A comparison with Kernelized Concept Erasure (Ravfogel et al., 2022) would be valuable, as both approaches perform null-space removal of protected-attribute information in a kernel/RKHS setting but rely on different mechanisms (ridge-based vs adversarial minimax game removal).
Ravfogel, Shauli, et al. "Adversarial concept erasure in kernel space." Proceedings of the 2022 Conference on Empirical Methods in Natural Language Processing. 2022.

**Strengths And Weaknesses:**

Fairness in the continuous setting is less studied than the discrete case. The authors generalize iterative null-space projections to kernel feature spaces, allowing fairness mitigation for non-linear models. Since the method transforms the kernel matrix itself, it is model-agnostic and can be used with standard algorithms like SVR or KRR without requiring changes to their training process. Empirical evaluations show competitive fairness–accuracy trade-offs on three real-world datasets.

---

> ### Author Rebuttal · Authors · 2026-03-29
>
> Dear reviewer, thank you for your suggestions and valuable feedback!
>
> **Comparison to Linear Removal**
>
> > How does the proposed kernelized approach compare empirically to a linear null-space projection baseline applied directly in the original feature space? Does using a kernel representation provide a clear advantage in reducing dependence on p while preserving predictive performance?
>
> Thank your pointing to this! The differences are dataset specific: if the dependency between p and the features is only linear it is fully sufficient to project in the original feature space.
> We run this experiment on the ACSIncome dataset, the results are available at this anonymous link: https://anonymous.4open.science/r/RebuttalICML-FF7C/Rebuttal_Linear_Comp.pdf.
>
> ''SVR-INPL'' denotes linear projection in the original feature space with a SVR applied afterwards to the projected features, otherwise using the same parameters (e.g. rbf kernel) as listed in the Appendix (and as ''SVR-FKD'') with the exception that we use $m\in \{0, 60, 100, 160, 180, 200\}$ for ''SVR-INPL'' and $m\in\{0, 45, 60, 80, 100, 140\}$ for ''SVR-FKD''.
>
> Evidently, at the beginning ''SVR-INPL'' is able to remove the linear dependencies even though the improvements in fairness for the same $m$ is much smaller than for ''SVR-FKD'', indicating that the amount of information removed about the protected attribute is smaller at each step for  ''SVR-INPL''. At a certain point the linear approach fails to capture the remaining non-linear dependencies which then be taken into account by the subsequent Kernel SVR. Thus ''SVR-INPL'' cannot further improve fairness (even with larger number of iterations e.g. 160, 180, 200) whereas our approach is able to do so.
> Interestingly, this effect is more pronounced for HGR and PF than for GDP.
>
> We will add this experiment to the camera-ready version.
>
> **Connection to [1]**
>
> > Given the connection to Kernelized Concept Erasure (Ravfogel et al., 2022), could the authors provide an empirical comparison with that method? In particular, how do they compare in terms of the fairness–accuracy trade-off?
>
> The focus of [1] and in the corresponding implementation is on categorical attributes and is thus not applicable to the setting of this work. However, it seems that a modification of [1] could be possible for continuous attributes which would be a very interesting endeavour for future work.
>
>
> [1] Ravfogel, Shauli, et al. "Adversarial concept erasure in kernel space." Proceedings of the 2022 Conference on Empirical Methods in Natural Language Processing. 2022.
>
> **On remaining dependencies**
>
> > The method does not guarantee independence from the protected attribute, as it only removes directions that are linearly predictive of p in the RKHS,so nonlinear dependencies in this representation may still remain in the final predictions.
>
> From the perspective of the RKHS this is true, so if we would use another non-linear approach inside the RKHS one could recover dependencies.
> Note that kernel methods correspond to a linear model inside that RKHS so they cannot take these non-linear dependencies into account for the predictions - i.e. as shown above we want to emphasize that our method is more powerful than an approach only linearly projecting in the original feature space.

---

> > ### Author Rebuttal · Reviewer_sv4m · 2026-04-03
> >
> > I thank the authors for the detailed rebuttal. The additional experiment with linear projection in the original feature space addresses my first question. Adding this baseline on other datasets could further help illustrate the main advantage of the method. I have therefore raised my score.
> >
> > My remaining concern is that this method does not directly optimize the fairness-accuracy trade-off and may discard information that is useful for predicting Y.

---

### Official Review · Reviewer_pUPf · 2026-03-12

**Soundness:** 3
**Presentation:** 2
**Significance:** 3
**Originality:** 2
**Overall Recommendation:** 4
**Confidence:** 3

**Summary:**

This paper introduces a null-space projection to achieve fairness for the nonlinear setting, more precisely for support vector regression. the method first use ridge regression to identify a projection which contains information about the protected attribute, and subsequently removing this with a null space projection, and subsequently deriving a kernel from the new space and then use this for downstream task analyses. The method is further motivated by the use of continuous protected attributes in contrast to the most common binary versions.

The method is compared to Perez-Suay et al's fair kernel ridge regression and a neural network with fairness penalty (Mary et al) across three benchmark datasets. In most cases the methods are comparable, but in some instances the proposed methods vastly outperforms the other methods.

The method is general and can be applied to any kernel based method.

**Compliance With Llm Reviewing Policy:**

Affirmed.

**Key Questions For Authors:**

Is it possible to include a study or reflection on limitations under label misspecifications?
The same for noise of the protected attribute, what is the influence?

**Limitations:**

The computational limitations of the method is described, but the aspects of performance unde noise and misspecification not.

**Strengths And Weaknesses:**

Strengths: The method is general and can be used in a variety of settings, although the core is not very novel, the use is.

Weaknesses: The method is compared to a limit subset of fairness methods, and limitations like behavior under label misspecification is not investigated.

---

> ### Author Rebuttal · Authors · 2026-03-29
>
> Thank you for your time taken to provide feedback and suggestions for improving our work!
>
> **Protected/Target Noise**
>
> > Is it possible to include a study or reflection on limitations under label misspecifications? The same for noise of the protected attribute, what is the influence?
>
> That is an interesting idea, especially for continuous protected attributes it now opens the door to analyze such influences more easily. On the one hand, the used datasets are collections of real-world data naturally already containing noise so our experiments suggest that we are robust against that. On the other hand, running a full analysis on this by manually adding noise in a controlled and/or structural fashion for the protected attribute as well as targets is a promising direction for a follow-up work, given the time constraints of the response phase.

---

### Official Review · Reviewer_Vd8h · 2026-03-12

**Soundness:** 4
**Presentation:** 4
**Significance:** 4
**Originality:** 4
**Overall Recommendation:** 5
**Confidence:** 3

**Summary:**

This paper addresses the under-explored problem of continuous fairness, i.e., fairness when the protected attributes are continuous rather than discrete. The authors extend the iterative null-space projection framework to kernel induced feature spaces. The key contribution is a derivation showing that null-space projections in the empirical feature space can be expressed as a direct, iterative transformation of the kernel matrix, thereby avoiding explicit computation in infinite-dimensional feature spaces. The authors propose a model-agnostic Fair Kernel Decomposition algorithm for the procedure. Empirical evaluation is conducted on Communities & Crimes and ACS dataset.

**Compliance With Llm Reviewing Policy:**

Affirmed.

**Key Questions For Authors:**

- The kernels are constructed on the original feature space before the FKD transformation procedure. Is is possible that the protected attributes are transform invariant such that it could not be fully removed by the null-space projection?
- Can you provide a theoretical guarantee on the convergence? More specifically, can you show some results showing how the information of protected attribute is related to the number of iterations $m$?

**Limitations:**

The authors have addressed the limitations in a dedicated section.

**Strengths And Weaknesses:**

**Strengths**:
-  The problem of handling continuous sensitive attributes is highly relevant to algorithmic fairness, and the paper provides clear motivation. The real-world scenarios involving age and proportion of a racial group as continuous sensitive attributes are practical and well-chosen, grounding the work in meaningful applications.
- The derivation of the kernel matrix transformation that corresponds to null-space projection in the empirical feature space is clean and mathematically sound.
- The evaluation of fairness spans both demographic parity and equalize odds. The authors also visualize the trade-off between utility and fairness violations. In addition, the author added the evaluation with multiple protected attributes.

**Weaknesses**:
- The algorithm requires inverting a matrix with the complexity of $O(n^3)$. The author also acknowledged this issue and only ran the experiment on the smaller Montana state for the ACS data. I don't think this algorithm can be easily applied to large-scaled dataset.
- There is no formal guarantee on the proposed method, like how many iterations are needed to achieve the the convergence?

---

> ### Author Rebuttal · Authors · 2026-03-29
>
> We thank you for your thorough review as well as for acknowledging our contributions!
>
> **Theoretical Guarantees**
>
> > Can you provide a theoretical guarantee on the convergence? More specifically, can you show some results showing how the information of protected attribute is related to the number of iterations $m$?
>
> Theoretically, the upper limit for $m$ is given by the number of training samples as otherwise all dimensions in the empirical feature space become uninformative.
> Apart from that this depends on how informative the features are of the protected attribute which is problem specific.
>
> **Invariant Features**
>
> > Is is possible that the protected attributes are transform invariant such that it could not be fully removed by the null-space projection?
>
> If the used Ridge Regression fails to capture the dependency this would be possible, for an artificial example one could use an extremely large value for the regularization parameter $\tilde{\alpha}$.
>
> **Scalability**
>
> See our comment for Reviewer frFe.

---

> > ### Author Rebuttal · Reviewer_Vd8h · 2026-04-03
> >
> > N/A

---

### Decision · Program_Chairs · 2026-04-30

**Decision:**

Accept (regular)

**Comment:**

The paper proposes an extension of iterative null-space projections to the kernel setting. The reviewers are overall positive and feel that this paper is a valuable contribution to the conference. Various concerns that were raised by the reviewers were addressed in the rebuttal. I therefore recommend to accept the paper. In terms of continuous fairness, there are kernel methods in the context of fairness which address this (e.g. Grunewalder and Khaleghi, Oblivious Data for Fairness with Kernels, JMLR, 2021). I would suggest to add a literature review instead of just stating that the literature is scarce.